# Young men's everyday life experiences with contraception and unintended pregnancy in Papua New Guinea

Stephen Bell[1,2]*, Elke Mitchell[2,3], Sophie Ase[4†], Herick Aeno[4], Richard Naketrumb[4], Priscilla Selon Ofi[4], Agnes Mek[4], William Pomat[2,4], Marie Habito[1,5], Glen Mola[6], Elissa Kennedy[1,5,7], Angela Kelly-Hanku[2,4]

1 Burnet Institute, Melbourne, Australia, 2 Kirby Institute, UNSW Sydney, Sydney, Australia, 3 Melbourne School of Population and Global Health, Melbourne University, Melbourne, Australia, 4 Papua New Guinea Institute of Medical Research, Goroka, Papua New Guinea, 5 Murdoch Children's Research Institute, Melbourne, Australia, 6 University of Papua New Guinea, Port Moresby, Papua New Guinea, 7 School of Public Health and Preventative Medicine, Monash University, Melbourne, Australia

† Deceased.
* Stephen.bell@burnet.edu.au

**Data Availability Statement:** The research data are confidential and are not publicly available. Excerpts of selected transcripts have been made available within the paper. Access to the de-identified

## Abstract

Unintended adolescent pregnancy is a public health priority in Papua New Guinea (PNG), where national policies specify need for easier access to reliable modern contraceptives. To reduce young people's experiences of unintended pregnancy in PNG, improved understandings of use of modern and other forms of contraception within young people's relationships is required to support the development of new sexual and reproductive health (SRH) programs and policies. The aim of this paper is to understand young men's use of modern and other contraceptives. This qualitative study involved semi-structured interviews with 35 sexually active young men aged 15–24 years, who were sampled purposively from the general population within community-based settings. Data were analysed using inductive thematic analysis techniques. Our analysis of young men's everyday experiences of using condoms and other contraceptives highlights clear drivers of unintended adolescent pregnancies. Across three settings, these included non-use of any modern method at first sex or during early sexual experiences; inconsistent use of condoms, often only after first pregnancy experiences; difficulties accessing condoms from health service providers, pharmacies and stores; a lack of understanding of other modern contraceptive strategies; inconsistent use of the withdrawal method; and inconsistent and incorrect use of calendar approaches due to misunderstandings about women's fertile period. Ten young men had never used any form of contraception. These occurred largely because young men's sexual agency is constrained within sexual and peer relationships, and community, school and health service settings, in ways that inhibit pregnancy prevention. It is important to engage meaningfully with young men to build sexual and reproductive health policies and programs that pay honest, respectful attention to young people's everyday sexual and social lives. Young men's everyday stories provide a unique lens through which we can identify mechanisms of change required to address the health and social

minimal dataset can be provided on reasonable request. Please direct requests for access to Mr Rodney Stewart (rodney.stewart@burnet.edu.au).

**Funding:** This work was supported by the National Health and Medical Research Council, Australia (Project Grant APP1144424 to SB). The funders had no role in study design, data collection and analysis, decision to publish, or preparation of the manuscript.

**Competing interests:** The authors have declared that no competing interests exist.

inequities associated with unintended pregnancy among young men and young women in PNG and beyond.

## Introduction

Unintended adolescent pregnancy is a global public health priority [1–3]. Maternal disorders are among the leading causes of death of girls aged 15–19 years in Asia and the Pacific [4]. Health outcomes include increased risk of maternal mortality and morbidity (e.g. postpartum bleeding, anaemia, pre-eclampsia, obstetric fistula, and complications with unsafe abortion) [1, 4], and poor perinatal outcomes (low-birth weight, preterm birth, perinatal mortality, neonatal mortality) [5]. Socio-economic outcomes (e.g. school drop-out, constrained livelihoods) perpetuate poverty and gender inequality through the life course and across generations [1, 6]. Unmet need for modern contraceptives among young women and young men is a key driver of unintended pregnancy in the region [7–9].

In Papua New Guinea (PNG), national policies specify need for easier access to adolescent-responsive health services and reliable modern contraceptives [10, 11]. However, in settings across PNG, young people's access to, uptake and (dis)continuation of contraceptives occurs in complex social contexts. These discourage sex before marriage, limit young people's access to information and services that support use of modern contraceptives in premarital relationships, constrain communication about sex and relationships and marginalise young women's decision-making power in intimate interpersonal heterosexual relationships [8, 12, 13]. Thus, while one in six young women aged 20–24 years have commenced childbearing by age 18 in a context of high maternal mortality [7], Demographic and Health Survey (DHS) data published in 2019 [14] indicate that less than half of females aged 15–24 years have demand for contraceptives satisfied by modern methods. Use of any contraception among unmarried sexually-active young people aged 15–19 years is very low (15%); over half who are using contraception rely on event-driven methods used by male partners (i.e. condoms, withdrawal) [14]. To reduce young people's experiences of unintended pregnancy in this setting, improved understandings of use of modern and other forms of contraception within young people's relationships is required to support the development of new sexual and reproductive health (SRH) programs and policies.

In 2018, we advocated youth-centred research–with young women and young men, aged 10–24 years in PNG and other LMICs–to inform the development of SRH policies, services and programs that pay honest, respectful attention to young people's everyday sexual and social lives [6]. Key to this call was engagement with young men on issues related to unintended pregnancy [6]. Through their partner becoming pregnant, young men experience pregnancy in ways that are social, with implications for both people's lives at point of pregnancy and thereafter. Understanding young men's limited, constrained and misuse of contraceptive strategies provides insight into the decisions and actions within young people's sexual relationships that lead to experiences of unintended pregnancy. Such insights can only be gained from research with young men, and are important in contexts like PNG where strong heteronormative masculinities mean that men tend to constrain women's sexual agency [15, 16], and social expectations constrain young people's access to modern contraception [8, 12, 13].

Drawing on the narratives of young men living in rural, peri-urban and urban areas of PNG, this paper aims to understand their use of modern and other contraceptives in the context of pregnancy prevention and unintended pregnancy. We use the term 'sexual agency' [17]

to refer to the strategies, actions and negotiations involved in young men's use of contraceptives and experiences of unintended pregnancy. We also draw on sociological [18] and anthropological [19] literature advocating attentiveness to 'everyday life' as a means to understand the socio-structural contexts within which young men and their sexual partners experience systematic 'structuring of vulnerability' [20] to unwanted SRH outcomes.

## Methods

This paper draws on a qualitative study involving 70 sexually active young people aged 15–24 years, recruited from the general population, to explore their experiences of pregnancy in PNG. Here we focus on the narratives of 35 young men. The study was informed by an interpretive approach to access locally-situated, subjective understandings of social life, recognising that people's understandings of the world are contextual and produced through experience and social interaction with others [6, 21].

Between June 2019 and April 2021, data collection was conducted in three settings: the capital city of Morobe Province; a small peri-urban settlement in the Western Highlands Province; a rural community in the Eastern Highlands Province. These were chosen due to the variability of SRH support services for young people in terms of health service structures and facilities, different models of service- and community-based service provision models, the range of provider types, and the variety of services available in any specific location.

Study participants self-reported experience of sexual intercourse and were sampled purposively [21] using sampling quotas to ensure balance with regard to gender and location, and a range of pregnancy experiences (i.e. never been pregnant; previous pregnancy experience, ending in either miscarriage, abortion or childbirth). Given the overwhelming lack of data from research with young people aged less than 18 years in PNG and other Pacific settings, we deliberately tried to recruit greater numbers of young people aged 15–18 years rather than 19–24 years.

In each setting, we had planned to recruit young people into the study using a variety of strategies: youth meetings held in local venues in each study location to introduce the study; distribution of flyers, posters and cards advertising the project, as appropriate in each setting. We were unable to use public recruitment approaches due to high sensitivity and stigma around premarital pregnancy. For this reason, young people were recruited into the study using more private snowball sampling strategies through local contacts in health services, non-governmental organisations and community settings that were established during prior research activities, personal contacts of researchers living in each area, and through early study participants.

A semi-structured interview guide was developed in English and translated into *Tok Pisin* by bi-lingual team members (see S1 File for English version). The final themes included community expectations of young people in relation to SRH; sexual practices and relationships; awareness and experiences of contraceptive strategies; experiences of pregnancy, giving birth, and parenthood; and experiences within health services. Interviews were conducted by trained Papua New Guinean social researchers in *Tok Pisin*, in audio-private settings perceived as safe by participants, at a time convenient to participants. Interviews lasted between 27–131 minutes, were audio-recorded, transcribed and translated into English and deidentified in preparation for thematic analysis.

Interview data were uploaded into NVivo X9 in preparation for inductive analysis, and were coded thematically following Strauss and Corbin's [22] system of 'open' and 'axial' coding. Open coding involves reading through the narrative data to increase familiarity with the material and to prepare 'theoretical memos' [22] as analytical reminders for generating ideas

and making links between different findings. Axial coding describes the later process of linking or organising open codes into themes and sub-themes, and providing evidence to support thematic findings [22].

Ethical approval was provided by the PNG Institute for Medical Research (PNGIMR) [#1807], the Government of PNG Medical Research Advisory Committee [#18.08] and UNSW Sydney [HC180555]. Pseudonyms are used to represent young men's voice. In line with guidance from PNG ethics committees, informed written consent was obtained from all participants; parental or guardian consent was waivered for those participants under the age of 18 who were considered liberated minors to the extent that they are making choices of an adult nature.

## Findings

Participant characteristics are described in Table 1. All 35 male participants reported being sexually active, and 23 had experiences of pregnancy; five described their sexual partner as pregnant at time of interview, 13 had experiences of induced abortion using medical or traditional strategies (two were unsuccessful), and 12 had children. Twenty-six young men were aware of condoms. None had heard of the emergency contraceptive pill, but five were also aware of contraceptive implants and injections, and the oral contraceptive pill. Awareness of condoms was higher among participants residing in urban or peri-urban areas. Awareness of condoms did not always lead to pregnancy prevention; of the 26 young men with awareness of condoms, 17 had experiences of pregnancy. Eleven young men, spread evenly across the three settings, reported ever having used condoms; seven had experiences of pregnancy. Nine young men–predominantly aged 15–18 years and living in rural areas–were not aware of, and had not used, condoms; six had experiences of pregnancy.

Participants described sexual experiences with young women of a similar age, including classmates, neighbours or young women met at community-based social or sports events. Sexual debut and experience were reported early, typically between 12–16 years. Most participants reported sexual experiences with multiple partners; some had current or previous experiences of concurrent sexual relationships. Sex was often reported as spontaneous and unplanned, sometimes occurring under the influence of alcohol.

### Not using condoms during early sexual experiences

Many young men described not knowing how to use condoms during their earliest sexual experiences, resulting in their sexual partners becoming pregnant quickly. Greg (21, peri-urban) had been in a four-year relationship with a girl during secondary school before they had sex for the first time. He explained that she became "pregnant the first time we had sex". Now married to the same girl, and with a child, Greg reflected on why this happened, noting the lack of education received on these topics at school. He said that at school, "we didn't receive any information about having sex with girls and how she would get pregnant and things like that. . . if I had known, I would have tried my best to avoid it; it's a shameful thing."

Joseph (18, peri-urban) started having sex with a girl he met at school when he was 15 years old. She became pregnant the second time they had sex. He described being "scared", left school and ran away from home while the girl decided to abort the pregnancy (strategy unknown) without his support. When asked whether he knew how to prevent pregnancy at this time, he said, "I had no idea. We were just [having sex], but it happened". He later learnt about condoms from friends though informal conversations:

**Table 1. Participant characteristics.**

| Participant characteristics | Male participants (n = 35) | | | |
|---|---|---|---|---|
| | **Urban** | **Peri-urban** | **Rural** | **Total** |
| Age | | | | |
| • 15–19 years | 5 | 10 | 10 | **25** |
| • 20–24 years | 6 | 2 | 2 | **10** |
| Married | | | | |
| • Yes | 1 | 2 | 2 | **5** |
| • No | 10 | 10 | 10 | **30** |
| School attendance | | | | |
| • In school | 6 | 5 | 3 | **14** |
| • Out of school | 5 | 7 | 9 | **21** |
| Pregnancy experiences | | | | |
| • Never pregnant | 4 | 4 | 4 | **12** |
| • Ever pregnant | 7 | 8 | 8 | **23** |
| Pregnancy outcomes | | | | |
| • Pregnant | 3 | 2 | - | **5** |
| • Miscarriage | - | - | - | - |
| • Attempted induced abortion | 1 | - | - | **2** |
| • Induced abortion | 2 | 3 | 6 | **11** |
| • With 1 child | 4 | 3 | 3 | **10** |
| • With 2+ children | - | 1 | 1 | **2** |
| Awareness of modern contraceptives | | | | |
| • None | - | 1 | 8 | **9** |
| • One type | 9 | 9 | 3 | **21** |
| • Two or more types | 2 | 2 | 1 | **5** |
| • Condom | 11 | 11 | 4 | **26** |
| • Contraceptive implant | 2 | - | - | **2** |
| • Oral contraceptive pill | - | 1 | 1 | **2** |
| • Emergency contraception | - | - | - | - |
| • Injection | 1 | 1 | - | **2** |
| • Vasectomy | - | 1 | 2 | **3** |
| Ever use of modern contraceptives | | | | |
| • None | 8 | 8 | 8 | **24** |
| • One type | 3 | 4 | 4 | **11** |
| • Two or more types | - | - | - | - |
| • Condom | 3 | 4 | 4 | **11** |
| • Contraceptive implant | - | - | - | - |
| • Oral contraceptive pill | - | - | - | - |
| • Emergency contraception | - | - | - | - |
| • Injection | - | - | - | - |
| Ever use of traditional and other strategies | | | | |
| • None | 3 | 6 | 7 | **16** |
| • One type | 7 | 5 | 4 | **16** |
| • Two or more types | 1 | 1 | 1 | **3** |
| • Withdrawal | 3 | 4 | 3 | **10** |
| • Rhythm/calendar method | 3 | 1 | 1 | **5** |
| • Traditional herbs | - | - | - | - |
| • Abstain | 3 | - | 2 | **5** |

I asked the boys. . . they said, 'Yeah you could [have prevented pregnancy] if you had used condoms but you did not so it's like that'. They talked about it and then I understood a little. . . we did not really talk about it much.

Simba (18, rural) had experienced two unintended pregnancies with one woman. He was not aware of pregnancy prevention strategies, and said, "for the second pregnancy I [started] thinking, 'Oh it's like this and that (had some ideas)'", but had not used condoms. From seeing information in hospitals he said, "I briefly understand that if you use condoms women will not fall pregnant", but explained, "I see that most [boys in the community] don't use condom nowadays."

## Not always using condoms

Inconsistent use of condoms was commonly reported. Phil (24, urban) said that when he was 16 years old, his girlfriend–a female classmate–became pregnant and they both left school. He "often had that thought" of pregnancy risks, had heard of condoms from male peers, but did not always use a condom when he drank alcohol at parties. Simon (18, peri-urban) had recently completed secondary school and reported being sexually active for the past four years. While none of the three girls he had sex with had become pregnant, he shared stories of other young men he knew who had experiences with unintended pregnancy, abortion and becoming parents younger than intended. He described thinking about preventing pregnancy, but said only that "I think about it myself" rather than discussing it with his sexual partners. He said, "once in a while or sometimes [I] use condom. When there is none [no condom], [I] just go without."

Some young men indicated that condoms were not always their first choice of contraception. Manda (16, rural) was in secondary school, had sex for the first time at 14 years and was having sex with three girlfriends at the time of interview. He experienced unintended pregnancy after having sex three times with one girl, which ended in an induced abortion using herbs. He described a mix of pregnancy prevention strategies which involved interpersonal conversations to assess risk with his girlfriends:

Before we had sex, I asked them 'is the date of your monthly period close by?'. . . they said 'no, my monthly period has already passed', and so we had unprotected sex a few times. The other time I used condom.

## Difficulties accessing condoms

Condoms were obtained from a range of providers, including "the hospital. . .[or] some stores and pharmacies" (Kerry, 24, urban) or "our clinic there, just nearby" (Joseph, 18, peri-urban). In addition to the health centre, aid post or community health worker, Manda (16, rural) noted how he relied on his friends, saying, "sometimes the boys have them in their house, so we normally get them and use them."

Despite some availability, a range of issues inhibited young men's access to condoms. For instance, the interviewer noted that Greg (21, peri-urban) lived close to the clinic, intimating that access to condoms should not be a problem. Greg pointed to the social concerns he had about accessing condoms from a public place, responding:

They keep the condoms at the top at the reception [on the counter] so the general public will see you going and collecting them. Just think about it. This is difficult and is a shameful thing, I was very ashamed to go and get it.

Several young men, especially in the rural area, noted difficult social interactions with adult health service providers when trying to access condoms. Manda (16, rural) said, "I go as a sick patient" to gain access to the doctor, who then "sometimes scolds me", questioning why he needs condoms. Manda explained, "I find it hard to say I want to do such [have sex]" but that the doctor advises him to be careful, as "you might accidently become a father." When asked about whether other young men access condoms like this, he said, "most times, they are often shy, so they just go and have sex without condoms."

## Reflections on other modern contraceptive strategies

Three young men who had used condoms were aware of, but not using, other forms of modern contraception, and awareness came from overhearing social conversations within community settings. Manda (16, rural) knew about condoms from older boys, and at school was taught about the oral contraceptive pill, as well as permanent methods such as vasectomy and tubal ligation.

Two married men were using condoms for child spacing and were considering other methods for the same purpose. After experiencing unintended pregnancy, Kerry (24, urban) realised the benefits of modern contraception, saying, "I think the best method is condom, and the second thing is going to family planning with your partner." Having heard about the contraceptive implant from a woman in his community, he confirmed his intention to go with his wife "for family planning to at least seek some advice from the doctors to avoid a second child." But his understanding of how these contraceptives work was limited, and were not the subject of conversations with his male friends.

Greg (21, urban) talked about using condoms for child spacing, now that he is a father. When asked about other forms of contraception, he explained his knowledge was limited, explaining, "I am shy, I don't talk a lot, so I do not ask." He had not sought information about other forms of modern contraception, but had heard that "women can go and receive injection or whatever to prevent pregnancy... I've heard of it, but my wife and I haven't tried it yet."

## Use of other contraceptive strategies

While not using condoms, 14 young men–in urban (n = 5), peri-urban (n = 6) and rural (n = 3) locations–used other strategies to prevent pregnancy. Ten had experiences of pregnancy.

**Withdrawal.**  Six young men used a withdrawal strategy based on advice from and conversation with peers; four had experiences of pregnancy. Pilan (18, rural), who had one child through an unintended pregnancy, said, "if you have sex with a girl or woman and you want to cum, you must not release your sperm inside her [vagina], you must just release outside." Chris (18, urban)–who reported sex with four girlfriends but had never experienced pregnancy–used this withdrawal technique as he was "scared" of unintended pregnancy. Krish (16, peri-urban) was sexually active with two girlfriends at the time of interview, had never experienced pregnancy, and was not worried about the risk of unintended pregnancy. He said, "these kinds of thoughts never come." Although aware of condoms, he had not used them before and instead used the withdrawal strategy based on advice from boys he knew.

**Calendar.**  The calendar method, also known as the rhythm method, was used by five young men, three of whom had experienced pregnancy. However, young men's narratives pointed to confusion about how to accurately identify the fertile period and when to avoid sex to prevent pregnancy, and misconceptions that women are all fertile at the same time. Terry (19, urban) had experienced sex only once with his current girlfriend. He had never

experienced pregnancy, and, relying on what he had learned in school, tried to assess risks of pregnancy in conversation with his girlfriend before they had sex:

> One time I was drunk and I'm a man too, so I went to her and asked her directly, I just asked her, 'I'd like to have now, so have you already had your monthly period or not?' Okay, I'm also a school student [educated] so I know the chart [menstrual cycle calendar] which we study in the Science and PD [Personal Development] lessons. Therefore, I went and asked her, and she said, 'I had it [period] already, it's okay now.' She said it was okay, so we had sex.

Mondo (17, rural) had experienced sex several times with one girl a year prior to the interview. She became pregnant and they aborted the pregnancy using herbs. He learned about the calendar method from his elder brothers who advised that before having sex he should try to ask his girlfriend questions to assess risk of pregnancy based on menstrual cycle. Mondo reported that his friends said,

> Ask her first if she is having her period or if it's close to the time to have her period. When you have sex with her and cum during those times, she will fall pregnant quickly. But if you ask her and she says no, then you two can have sex and cum inside her too but she won't fall pregnant. . . you as a man must ask her first if it's okay, and then later you two can have sex.

Luke (20, rural) described advice he had received from his grandfather, who told him, "look at the moon. When there is something circling the moon, that is not the right time to sleep with women, they will fall pregnant quickly."

**Abstaining from sex.**   Due to previous experiences of unintended pregnancy, four young men described abstaining from any sexual intercourse to avoid pregnancies. Mondo (17, rural), after his experience of induced abortion, said, "I always think twice about it. I don't want girls to become pregnant, only when I get married." He explained, "I made the other girl pregnant, and this has made me think about it and I'm a little scared about it." Kelly (18, urban) had been in a relationship with a female classmate for two years, and she became pregnant the first time they had sex. Having been "shocked" as it was his sexual debut, he had since decided to abstain from sex:

> It changed some of my behaviours towards having sex with girls. Since last year, the whole of last year up until now, I have not had sex with a girl. My two girlfriends, I just make friends with them. We just go around, chat, share things and we stay but I haven't had sex with either of them. . . I personally made up my mind not to have sex with girls until and unless I go to school and complete my education.

## No strategies used to prevent pregnancy

Ten young men–in urban (n = 3), peri-urban (n = 2) and rural (n = 5) locations–had never used a contraceptive strategy; six had experiences of pregnancy. These experiences point to the continued lack of support and conversation about these issues in social settings, as well as in school and health service settings.

Some young men explained that they had no idea about how women became pregnant or how to prevent pregnancy. Tommy (22, urban), who had experienced pregnancy twice, said, "I

did not know of any method to prevent a girl from being pregnant. . . I do not have any idea about that. I am still too young."

Sam (23, peri-urban) reported sexual debut when he was ten years old in primary school. Since then he had three experiences of pregnancy, and offered a longer-term perspective on his lack of knowledge about contraception. When he first experienced pregnancy at 18 years, he said he had no knowledge of pregnancy prevention, explaining, "I wasn't really mature. . . We were going around (having sex) and the woman somehow fell pregnant and I was also confused. 'Who made you pregnant?' I asked and she said, 'You!'." At the time of interview, Sam was married to a different woman with whom he had a son when he was 21 years old. She was also pregnant at the time of interview. He and his wife "have no idea" about preventing pregnancy, and use "nothing" to prevent pregnancy. He wants many children so contraceptives were not needed.

In the rural location, four young men lived up to eight hours' walk away from their local health post, school and community centre and seemed particularly vulnerable to unintended pregnancy. Blaze (15), Malik (15), Wilson (16) and Boa (15) all had multiple girlfriends with whom they were sexually active at time of interview. They had not heard of, and therefore not used, any contraceptive strategies. At the time of interview, Malik was having sex with two girlfriends, but neither had become pregnant. He knew of herbal strategies for terminating a pregnancy, but had no knowledge of how to prevent a pregnancy. Wilson described having had sex with "around 12" young women, one of which had led to an unintended pregnancy that ended in self-managed induced abortion (strategy unknown). When asked whether he was afraid that girls he was having sex with might fall pregnant, he replied "No, I normally go and have sex with them". He had not heard of any ways to prevent pregnancy, including not having heard of condoms, when specifically asked about them.

## Discussion

Our analysis of young men's everyday experiences of using condoms and other contraceptives highlights eight drivers of unintended pregnancy within their sexual relationships. These included: a lack of knowledge about condoms and other contraceptive strategies; non-use of any modern method at first sex; learning of condoms or other modern contraceptives after sexual debut or first pregnancy; a reliance on male condoms or other methods (withdrawal or calendar approaches) which are not used consistently or correctly; a preference for less effective, often misunderstood strategies such as withdrawal or calendar approaches; limited knowledge about reproduction and contraception; misunderstandings about women's fertile period and appropriate use of the calendar method; and poor access to condoms and other modern methods. However, these occurred largely because young men's sexual agency [17] was constrained within sexual and peer relationships, and community, school and health service settings, in ways that inhibit effective, consistent pregnancy prevention.

### Social support within peer networks

A strong theme running through young men's narratives was their reliance on other men's knowledge of contraceptives as a basis for learning. This is an issue reported in PNG [23] and other Pacific settings [9]. Young men were almost entirely reliant on established relationships of trust with other young men for information about contraceptives and pregnancy prevention, irrespective of how informed this advice was, and how late this advice was received. For instance, Simon (18, peri-urban) and Manda (16, rural) only knew about condoms based on conversations with from male peers. Joseph (18, peri-urban) described learning about condoms for the first time from informal conversation from friends, but only after he had already

experienced an unintended pregnancy. Male peer networks were sources of advice about other contraceptive strategies. For example, for Krish (16, peri-urban) about using a withdrawal strategy, and for Mondo (17, rural) who learned about the calendar method from his elder brothers. Beyond sharing of information, Manda (16, rural) also described how young men sometimes rely on their peer networks to access condoms for pregnancy prevention. These findings point to the opportunities of tapping into existing, trusted relationships young men have with their peers to enhance their SRH and prevent pregnancies through sharing of information and condoms in urban and rural settings in PNG.

### In conversation with sexual partners

The social dimensions of young men's relationships with sexual partners also influenced use of contraceptives. For example, some young men noted a lack of communication with their sexual partners about pregnancy prevention strategies. Simon (18, peri-urban) explained that he felt concerned about the risk of pregnancy occurring from having sex without a condom. However he did not communicate these concerns, nor decide on strategies to avoid these risks, with his girlfriend.

However, this was not always the case. Some young men described communication attempts with their sexual partners to assess risk and prevent pregnancy at point of sexual intercourse. For instance, Terry and Mondo described clear efforts to engage their sexual partners in quite open and frank discussion about the calendar method. They tried to ascertain whether it was safe to have sex without a condom based on their sexual partner's understanding of the timing of their menstrual cycles. Though undermined by young people's misunderstandings and inexperience with the calendar approach, this does point to young men and young women acting on information in ways to assess risk and prevent pregnancy. It also points to the possibility of successful communication and negotiation between young sexual partners at point of sexual intercourse to prevent pregnancy, if young men are fully armed with the skills, knowledge, know-how and contraceptive products upon which to base these risk assessments.

### Constrained community and institutional contexts

Young men's narratives point to the constrained support they receive within community, health service and school settings. This is predominantly derived from societal expectations that sex should not occur before marriage [8, 12, 13].

At community level, gender- and age-based norms and relations framing communication about SRH influenced use of contraceptive strategies in young men's sexual relationships. There was a lack of support reported from informed adults in their localities. When intergenerational communication was reported, this included family and community members sharing knowledge gained from their own lived experiences [23]. This tended to focus on withdrawal and calendar methods which are strategies they had used themselves.

Only three participants had heard of other long-acting reversible contraception such as implants or injections–one young man during school-based personal development lessons, and two married men from overhearing women's conversations in community settings. With such limited knowledge, young men in this study were not in a position to be able to discuss these options with sexual partners, nor advocate or support young women's access to or use of these other forms of modern contraceptives to prevent pregnancy, if they wanted to. Furthermore, narratives from married young men in the study illustrate the complexity of young men's engagement in decision making around use of modern contraceptives even after years of sexual experience [13]. For instance, Kerry and Greg pointed to the limited conversations

between them and their wives about modern contraception, and the notion that these are not conversations men are typically involved in Papua New Guinean communities [13]. Investing in efforts to build young people's knowledge and use of modern contraceptives, and their ability to communicate more openly about related SRH issues is important for current and future generations. Such investment will hopefully lead to a generational change in social support networks within community settings as young people grow up, become parents and have families themselves.

Barriers to sharing of information and provision of contraceptives from trained health providers were reported by young men. Teacher-based education in schools tended to be restricted to limited biological information about reproduction [24]. Young men reported that condoms were locally available. But their access was constrained by feelings of shyness and shame when trying to obtain them in health care and other public settings, and difficult, judgemental interactions with health service providers. Furthermore, echoing terminology used within the health system in PNG, young men used the term 'family planning' to describe services where modern contraceptives are provided to women after birth of their first child. Use of this terminology is problematic. It orients the purpose of contraception to child spacing which is perceived as relevant for people who are married, limiting unmarried people's access to services that would support prevention of unintended pregnancies.

## Reflections on study design

The sample of young men in this qualitative study was large enough to document a range of lived experiences related to use of condoms and other contraceptives, and unintended pregnancy, across three diverse settings in PNG. However, this was a qualitative study and care must be taken not to assume that this represents all young men's experiences of contraception and pregnancy beyond these three settings. Data collection by a team of adult interviewers may also have increased variation among individual responses. 'Internal reliability'[21], always a concern in rigorous qualitative research, was enhanced in several ways: the team of Papua New Guinean social researchers are well trained, with wide-ranging experiences of undertaking in-depth research with hard-to-engage populations about sensitive research topics; data was collected in local languages and during sustained periods of immersive fieldwork in each setting, which enhanced local understanding of issues and built trusted relationships with young study participants; interviewers worked together with study investigators throughout the study to ensure rigour and consistency in data collection and interpretation.

## Implications for policy and practice

Our analysis points to the systemic structuring [20] of young Papua New Guinean men's vulnerability to unintended pregnancy, starting from often young ages of sexual debut and lasting into marriage. In PNG, this vulnerability arises from both a lack of access to contraceptives and SRH services, and a lack of information and education provision in school, health service and community settings. A range of mechanisms to enhance men's sexual agency to reduce young people's vulnerability to unintended pregnancy in PNG can be identified from these findings. First, it is worth tapping into the established networks of trust and communication that exist between young men, by using peer-led strategies to improve access to, and acceptance of, condoms. Enabling young men to promote and distribute condoms–with the support of other community-located health structures that exist, such as village health volunteers– would be a potential strategy to overcome access barriers that exist due to geography and provision through formal adult-controlled health services. Second, supporting young Papua New Guinean men to learn about pregnancy and pregnancy prevention is key to enabling them to

work with their sexual partners to prevent unintended pregnancies. Topics would include a fuller range of modern contraceptives (emergency contraception, oral contraceptive pills, implants and injections), and clarifying misconceptions about reproduction and calendar methods. Working with young people to co-design catchy, memorable and accurate bite-sized messages in *Tok Pisin* and other local languages, that can be shared safely and widely through peer and other local networks, would be an innovative starting point. Third, use of role modelling of young men's sexual agency could help young men negotiate safe sexual experiences and tackle gender norms and relations. Examples might include the narratives of people like Terry and Mondo, and Mondo's statement, "you as a man must ask her first if it's okay, and then later you two can have sex". Finally, young men and young women in PNG require pragmatic access to contraceptive services offering an expanded range of safe, modern contraceptives (including emergency contraception) at an earlier age, in time for their first sexual experiences, rather than just child spacing processes in marriage. This will require advocacy at government, health system and health service levels to enhance young women's access to contraceptive services under the age of 16 years without parental consent [11], as well as throughout community settings to enable local support for young people's access to these services.

## Conclusions

We close with a final note on the value of attentiveness to everyday life and lived experience. People in positions of power are increasingly expected to listen to or work with people with 'lived experience' of inequity or injustice associated with a particular identity, health condition or event, in this case pregnancy. Knowledge and expertise gained from firsthand experience is distinct from other more recognised forms of expertise and authority, held by policy makers and medical professionals, for example. While lived experience-centred policy and practice is now widely accepted in mental health [25] or HIV [26], the 'battle for credibility' [27]–about who can speak effectively and authoritatively about a health issue, and which voices are heard and respected–has barely started in relation to unintended adolescent pregnancy, encumbered by prejudice around the perceived knowledges, capabilities and expertise of young people.

Our analysis contributes to the important but limited qualitative data on young people's experiences of pregnancy and pregnancy prevention strategies in PNG [6, 8, 23]. Though rarely involved in reproductive health studies, or the design of SRH policies and programs, young men's everyday stories provide a unique lens through which we can identify mechanisms of change required to address the health and social inequities associated with unintended pregnancy among young men and young women in PNG and beyond [6]. These findings suggest the need for targeted interventions that address the specific barriers young men face in accessing and using contraceptives effectively.

## Supporting information

**S1 File. Interview guide.**
(DOCX)

**S1 Checklist. Inclusivity in global research.**
(DOCX)

## Acknowledgments

Sophie Ase passed away before the submission of the final version of this manuscript. Stephen Bell accepts responsibility for the integrity and validity of the data collected and analysed. We dedicate this paper to her memory.

## Author Contributions

**Conceptualization:** Stephen Bell, William Pomat, Glen Mola, Elissa Kennedy, Angela Kelly-Hanku.

**Data curation:** Stephen Bell, Elke Mitchell, Sophie Ase, Herick Aeno, Richard Naketrumb, Priscilla Selon Ofi, Agnes Mek.

**Formal analysis:** Stephen Bell, Elke Mitchell, Marie Habito.

**Funding acquisition:** Stephen Bell, William Pomat, Glen Mola, Elissa Kennedy, Angela Kelly-Hanku.

**Investigation:** Stephen Bell, Elke Mitchell, Sophie Ase.

**Methodology:** Stephen Bell, Elke Mitchell, Sophie Ase, Herick Aeno, Richard Naketrumb, Priscilla Selon Ofi, Agnes Mek, William Pomat, Glen Mola, Elissa Kennedy, Angela Kelly-Hanku.

**Project administration:** Stephen Bell.

**Supervision:** Stephen Bell, Elke Mitchell, Sophie Ase, Herick Aeno, Angela Kelly-Hanku.

**Writing – original draft:** Stephen Bell.

**Writing – review & editing:** Stephen Bell, Elke Mitchell, Herick Aeno, Richard Naketrumb, Priscilla Selon Ofi, Agnes Mek, William Pomat, Marie Habito, Glen Mola, Elissa Kennedy, Angela Kelly-Hanku.

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
