## [Decision Letter · Decision Letter 0]

26 Jun 2024

PGPH-D-24-01087

Young men’s everyday life experiences with contraception and unintended pregnancy in rural, peri-urban and urban settings in Papua New Guinea

Dear Dr. Bell,

Thank you for submitting your manuscript to PLOS Global Public Health. After careful consideration, we feel that it has merit but does not fully meet PLOS Global Public Health’s publication criteria as it currently stands. Therefore, we invite you to submit a revised version of the manuscript that addresses the points raised during the review process.

Please note that we have only been able to secure a single reviewer to assess your manuscript. We are issuing a decision on your manuscript at this point to prevent further delays in the evaluation of your manuscript. Please be aware that the editor who handles your revised manuscript might find it necessary to invite additional reviewers to assess this work once the revised manuscript is submitted. However, we will aim to proceed on the basis of this single review if possible. 

The reviewers comments focus on the reporting of the methodology specifically the sample size calculations as well as providing suggestions on how to evaluate the results in more depth. Their comments are available below. Please review and address these comments in your revisions

We look forward to receiving your revised manuscript.

Kind regards,

Emma Campbell, Ph.D

Staff Editor

Journal Requirements:

Additional Editor Comments (if provided):

Reviewers' comments:

Reviewer's Responses to Questions

**Comments to the Author**

1. Does this manuscript meet PLOS Global Public Health’s publication criteria? Is the manuscript technically sound, and do the data support the conclusions? The manuscript must describe methodologically and ethically rigorous research with conclusions that are appropriately drawn based on the data presented.

Reviewer #1: Partly

2. Has the statistical analysis been performed appropriately and rigorously?

Reviewer #1: N/A

3. Have the authors made all data underlying the findings in their manuscript fully available (please refer to the Data Availability Statement at the start of the manuscript PDF file)?

Reviewer #1: No

4. Is the manuscript presented in an intelligible fashion and written in standard English?

Reviewer #1: Yes

5. Review Comments to the Author

Reviewer #1: Overall

This paper provides findings from an interesting, large (for qualitative) sample of sexually active men in Papua New Guinea. The introduction promises interesting inquiries into topics and contexts that do not end up being presented in the paper. Given the size of the sample and the purported goal of the research, too much is left unsaid.

The main conclusions presented here do not break new ground, nor do they highlight the specific contexts that are alluded to on line 89 (“complex social, cultural and religious contexts.”) The introduction refers to the way young men relate to pregnancies in ways that are “social, relational, gendered, and emotional” (line 98) but the main themes introduced do not get there either. The paper does not identify themes that lead to “mechanisms for change” (line 113) alluded to in the Introduction. Insufficient information is provided about the data and the data collection.

Abstract

Provide information on the sample (where they were recruited from).

There seems to be some circular logic in the major themes listed in the abstract here – low contraceptive uptake is defined by, or precedes, non-use of modern contraceptive methods, and low use, or inconsistent use of condoms, withdrawal, and calendar methods.

Introduction

Compared to US rates (77% 2006-2010, per NCHS) , the % of births among age 15-19 that are reported as unintended here (“more than a quarter,” line 83 – which I guess covers everything from 26 through 100%, but is misleading) is markedly low. If this were in fact the rate, how does this highlight the public health priority for reducing unintended births? How does this relate to the complex social, cultural, and religious contexts (line 89).

In the title, abstract, and the introduction it should be made clear that this is a sample of men from the general population – I think? As noted below, there is inadequate information on from where participants were recruited or sampled and wh.

While lofty, the claims lines 111-113 that the authors “take a deliberate ethical (18) or values-based (21) stance, located in the desire for social justice – we centre, work with and learn from the stories of young men to identify mechanisms of change required to address inequities that undermine the health and wellbeing of young people” would need to be presented in the Discussion, not the Introduction, and would need to be supported with data and / or methods.

Methods

How were the participants recruited and sampled (beyond purposeful sampling using snowball sampling). Why snowball sampling (especially given the claim, line 435-6 a range of lived experiences)? Provide some rationale for why this method was used – presumably it would not be that hard to find sexually active young men who don’t represent a specific population.

Findings

As mentioned above – the intro and discussion talk about nuanced concepts (sexual debut, for example) that are not captured in the themes, which seem very one-dimensional (e.g. “not using condoms during early sexual experiences.”)

Line 190-192 presents a very rich example – where the respondent discusses having sex with a partner who is already pregnant – is the participant indicating that his understanding is that she got pregnant through sexual activity that he does not consider having “sex”?

Difficulties accessing condoms, for instance – this could intersect quite a bit with social and cultural and religious themes that were referred to in the Intro, but it does not delve into any depth.

Given the amount of data, tables could be used to present more data with more in-depth exploration of the quotes.

The data collection instrument/interview guide should be provided as an appendix.

Implications for Policy and Practice

The implications follow from the findings, but they don’t bring into the discussion the specific context of PNG.

6. PLOS authors have the option to publish the peer review history of their article (what does this mean?). If published, this will include your full peer review and any attached files.

**Do you want your identity to be public for this peer review?** For information about this choice, including consent withdrawal, please see our Privacy Policy.

Reviewer #1: No

---

## [Decision Letter · Decision Letter 1]

11 Oct 2024

PGPH-D-24-01087R1

Young men’s everyday life experiences with contraception and unintended pregnancy in rural, peri-urban and urban settings in Papua New Guinea

Dear Dr. Bell,

Thank you for submitting your manuscript to PLOS Global Public Health. After careful consideration, we feel that it has merit but does not fully meet PLOS Global Public Health’s publication criteria as it currently stands. Therefore, we invite you to submit a revised version of the manuscript that addresses the points raised during the review process.

The revised manuscript has been re-assessed by the reviewers. Reviewer  2 has provided some additional comments that require revisions in the attached document.  Please review these comments and make the appropriate changes to the manuscript.

Additionally, please include a detailed discussion of the 5 other publications that are related to this manuscript with citations where possible. In this discussion please include a scientific rationale for separating the study outputs into 6 manuscripts and detail how they differ.

We look forward to receiving your revised manuscript.

Kind regards,

Emma Campbell, Ph.D

Staff Editor

Journal Requirements:

Additional Editor Comments (if provided):

Reviewers' comments:

Reviewer's Responses to Questions

**Comments to the Author**

1. If the authors have adequately addressed your comments raised in a previous round of review and you feel that this manuscript is now acceptable for publication, you may indicate that here to bypass the “Comments to the Author” section, enter your conflict of interest statement in the “Confidential to Editor” section, and submit your "Accept" recommendation.

Reviewer #1: All comments have been addressed

Reviewer #2: (No Response)

2. Does this manuscript meet PLOS Global Public Health’s publication criteria? Is the manuscript technically sound, and do the data support the conclusions? The manuscript must describe methodologically and ethically rigorous research with conclusions that are appropriately drawn based on the data presented.

Reviewer #1: Yes

Reviewer #2: Yes

3. Has the statistical analysis been performed appropriately and rigorously?

Reviewer #1: N/A

Reviewer #2: Yes

4. Have the authors made all data underlying the findings in their manuscript fully available (please refer to the Data Availability Statement at the start of the manuscript PDF file)?

Reviewer #1: No

Reviewer #2: Yes

5. Is the manuscript presented in an intelligible fashion and written in standard English?

Reviewer #1: Yes

Reviewer #2: Yes

6. Review Comments to the Author

Reviewer #1: The revisions made were responsive, thank you.

Reviewer #2: (No Response)

7. PLOS authors have the option to publish the peer review history of their article (what does this mean?). If published, this will include your full peer review and any attached files.

**Do you want your identity to be public for this peer review?** For information about this choice, including consent withdrawal, please see our Privacy Policy.

Reviewer #1: No

Reviewer #2: No

---

## [Editor Report · Decision Letter 2]

21 Oct 2024

Young men’s everyday life experiences with contraception and unintended pregnancy in Papua New Guinea

PGPH-D-24-01087R2

Dear Dr Bell,

We are pleased to inform you that your manuscript 'Young men’s everyday life experiences with contraception and unintended pregnancy in Papua New Guinea' has been provisionally accepted for publication in PLOS Global Public Health.

Best regards,

Julia Robinson

Executive Editor